# Social inequalities in the misbelief of chloroquine's protective effect against COVID-19: results from the EPICOVID-19 study in Brazil

**Bruno Pereira Nunes** [1,2*], **Inácio Crochemore-Silva** [3,4], **Grégore I. Mielke** [5],
**Luis Paulo Vidaletti** [3], **Marilia Arndt Mesenburg** [6,7], **Mariangela Freitas da Silveira** [3],
**Pedro C. Hallal** [1,3]

1 Department of Health and Kinesiology, University of Illinois Urbana-Champaign, Urbana, Illinois, United States of America, 2 Postgraduate Program in Nursing, Universidade Federal de Pelotas, Pelotas, Brazil, 3 Postgraduate Program in Epidemiology, Universidade Federal de Pelotas, Pelotas, Brazil, 4 Postgraduate Program in Physical Activity, Universidade Federal de Pelotas, Pelotas, Brazil, 5 School of Public Health, The University of Queensland, Brisbane, Australia, 6 Postgraduate Programme in Epidemiology for Public Health, Pontificia Universidad Católica del Ecuador, Quito, Ecuador, 7 Fernandes Figueira National Institute of Women, Children and Adolescents Health Fundação Oswaldo Cruz, Rio de Janeiro, Brazil

* nunesb@illinois.edu

## Abstract

### Objectives

This study aimed to assess the dissemination of anti-science messages regarding COVID-19 in Brazil, specifically examining how social inequalities contributed to the misconception that chloroquine has a protective effect against the virus.

### Study design

Three countrywide population-based studies were conducted in 2020 (May 14–21, June 4–7, and June 21–24), including 133 Brazilian cities (N = 74,077).

### Methods

Participants (≥20 years old) were asked whether they believed in chloroquine's protective effect against infection with the SARS-CoV-2 virus (no/yes/don't know). "Yes" and "don't know" answers were considered misconceptions (effect of denialism). A jeopardy index score was calculated to assess cumulative social deprivation based on sex, race and ethnicity, and socioeconomic variables. Descriptive analysis and inequality measures (Slope Index of Inequality – SII; and Concentration Index) were used to evaluate the association between believing in chloroquine's protective effect against COVID-19 and the jeopardy index. Multinomial logistic regression was used in the unadjusted and multivariable analysis.

**Data availability statement:** The EPICOVID19 datasets are freely available online at https://epidemio-ufpel.org.br/epicovid/. All other relevant data are within the paper and its Supporting information files.

**Funding:** The study was funded by the Brazilian Ministry of Health (TED 012/2020 SAPS), Instituto Serrapilheira (2980-7), Brazilian Collective Health Association (S/N), and JBS Fazer o Bem Faz Bem (S/N). The funders have no role in the study design, data collection and analysis, interpretation of findings, or manuscript writing.

**Competing interests:** The authors have declared that no competing interests exist.

## Results

Overall, 47.6% of participants either believed that chloroquine prevented COVID-19 or stated, "I don't know." Marginalized racial and ethnicity groups, those with low education level, and those with low socioeconomic status were more likely to erroneously believe that chloroquine prevented COVID-19. The chance of lack of knowledge (Don't know) was higher (Odds Ratio: 2.57,CI95% 2.21; 2.99) among women, Black/Brown/East Asian/Indigenous, and among those in the lowest education level and wealth quartiles compared to men, white individuals, and those in the highest education and wealth quartiles. Absolute and relative inequalities were observed according to the jeopardy index. The highest absolute inequality was observed for the category "I don't know" (SII = −15.1).

## Conclusions

Misbelief in chloroquine's protective effect against SARS-CoV-2 was high in Brazil. People with greater social vulnerability were more likely to wrongly believe that chloroquine prevented COVID-19.

## Introduction

The COVID-19 pandemic has had several negative consequences for the global population [1,2]. The most striking impact was the high avoidable mortality caused by inadequate management of the SARS-CoV-2 virus spread. Brazil holds the second-highest number of confirmed COVID-19 deaths (702,421; May 2023), with a mortality rate of over 330 deaths per 100,000 inhabitants. Despite representing 2.7% of the world population, the country accounts for 10.5% of the COVID-19 mortality as of September 2022 [3]. In addition, mortality was unequally distributed in the population [4–6]. The higher mortality among the most vulnerable individuals also worsened the ability of families to maintain and guarantee a family budget sufficient for dignified survival, increasing, for example, food insecurity [7]. In addition to population health issues, there was a deepening of social inequalities [8,9].

Inadequate pandemic management is related to scientific denialism [10], as observed in Brazil [11,12]. Throughout the pandemic, the former Brazilian president (in office 2019−2022) undertook actions that undermined science, including disputes with state governors over political party affiliations, the relativization of the magnitude and severity of the SARS-CoV-2 virus, and the creation of a dichotomy between public health and economics. Even more severe was the strategy of denying scientific consensus, the delay in purchasing vaccines, and the defense of medicines known to be ineffective for preventing and treating infection, such as chloroquine [13].

As a result, there was a strong emphasis on chloroquine as an effective treatment for SARS-CoV-2 infection, leading to an indiscriminate use of prescription by doctors. Unfortunately, the Federal Council of Medicine failed to ban off-label use of

chloroquine for COVID-19. Such strategies and actions influenced the population's perception of the protective effect of chloroquine in preventing and treating SARS-CoV-2 infection [11].

The complexity of information dissemination in society suggests that the effect and dissemination of fake news may not be limited to electorally aligned individuals [4]. Inequitable access to high-quality information, limited access to health services, and increased difficulties in daily life can exacerbate the effects of the spread of fake news about chloroquine, making it more widespread and sustained among the most vulnerable populations.

In this context, jeopardy analysis, based on the intersectionality theory [14], is a relevant approach for evaluating epidemiological health indicators to measure social inequity [15–18]. Earlier evidence suggests greater SARS-CoV-2 virus infection and mortality in contexts marked by inequalities [4,5,19], especially among Black, mixed, or Indigenous individuals and those with lower education and household asset levels [5,6,20,21]. The systemic effect of scientific denialism in Brazil appears to have a widespread impact on society as found for mortality [4,22]. Moreover, opinion surveys suggest that the favorable perception of chloroquine is more prevalent among individuals with lower educational levels [23,24].

This study aimed to assess the spread of denialist messages regarding COVID-19 in Brazil, specifically examining how social inequalities contributed to the misconception that chloroquine has a protective effect against the virus. We hypothesized that scientific denialism is more prevalent among individuals with greater social vulnerability.

## Materials and methods

Data are from the EPICOVID-19 Brazil study, which includes three repeated seroprevalence studies conducted in 2020: 1) May 14th to 21st (n = 25,025); 2) June 4th to 7th (n = 31,165); and 3) June 21st to 24th (n = 33,207). Of these, 77,102 participants were 20 years or older. For each face-to-face survey, a three-stage probabilistic sample was selected (cities, urban census tracts, and households). Participants were tested for antibodies against SARS-CoV-2 using the WONDFO SARS-CoV-2 Antibody Test (Wondfo Biotech Co., Guangzhou, China), which detects the presence of antibodies against SARS-CoV-2 (IgG and IgM). Further methodological details are available elsewhere [20,25,26]. For the present study, 74,077 individuals aged ≥20 years or more with valid information for beliefs about COVID-19 and the main exposure (jeopardy index) were included in the analysis. This study complied with all ethical principles and legislation governing research involving human subjects and was approved by the National Research Ethics Commission (CAAE 30721520.7.1001.5313). All participants signed a written informed consent form.

The primary outcome assessed in this study was the percentage of the population holding the misconception that chloroquine protects against the SARS-CoV-2 virus. This was measured using the following question: "What do you believe offers protection against the coronavirus? [take chloroquine – no/yes/do not know]." In our study, responses of "yes" and "I do not know" were considered indicators of denialism during the pandemic due to the extensive dissemination of fake news and misinformation fueled by the President's actions and speeches regarding the "efficacy" of chloroquine in preventing and treating COVID-19.

A jeopardy index score was calculated based on the aggregation of four sociodemographic variables that represent dimensions of social privilege: biological sex (male; female), self-reported race and ethnicity in five groups ("White" *[Branco]*; "Brown" *[Pardo]*; "Black" *[Preto]*; "East Asian" *[Amarelo]*; and "Indigenous" [Indígena]) following the official Brazilian classification of ethnicity, education level (incomplete elementary school; complete elementary school; complete high school; complete university degree); and wealth score (divided into quartiles), which was based on characteristics and assets of the household [27]. A composite jeopardy index score was created by assigning the most privileged group for each variable a score of zero (men, white, highest education, and highest socioeconomic position) and the least privileged group a score of one (women and Black-Brown-East Asian-Indigenous) or three (none or incomplete primary level of education and the lowest quartile of socioeconomic position). Therefore, for each variable, the following scores were assigned: sex (men = 0; women = 1); race and ethnicity (White = 0; Black-Brown-East Asian-Indigenous = 1); education (university graduate = 0; complete secondary or incomplete university = 1; complete primary or incomplete secondary = 2; none or incomplete primary = 3); socioeconomic



position (top quartile = 0; 3rd quartile = 1; 2nd quartile = 2; bottom quartile = 3). The scores for each indicator were summed, resulting in a 'Jeopardy Index' ranging from 0 to 8. The index is lower for individuals with greater social privilege (or greater guarantee of human rights), which can also be interpreted as lower social vulnerability.

We conducted a descriptive analysis using prevalence and respective 95% confidence intervals (CI). The primary outcome ("believe in the protective effect of chloroquine against the SARS-CoV-2 virus") was assessed based on the primary exposure (jeopardy index) and all the variables that comprised the index. We employed multinomial logistic regression models to evaluate the crude and adjusted odds ratios of belief (yes) and lack of knowledge (don't know) regarding the chloroquine's protective effect against COVID-19 according to each independent variable (sex, race and ethnicity, education level, and wealth), with all variables mutually considered in the adjusted model. Based on the jeopardy index, we evaluated the crude effect of belief (yes) and lack of knowledge (don't know) regarding the protective effect of chloroquine against COVID-19.

Complex measures of inequality were calculated by using the Slope Index of Inequality (SII) and Concentration Index (CIX), with the jeopardy index as the exposure variable. The SII indicates the absolute difference in predicted values through logistic regression according to the jeopardy index. Therefore, it represents the absolute difference, in percentage points, between the estimated values for the extreme groups of the stratification variable. The deviation from linearity was assessed visually. The CIX assesses relative inequality, similar to the GINI index. Both indicators account for the entire distribution of the stratifier (jeopardy index) and are presented with values ranging from −100 to +100. Negative values reveal more pronounced inequalities among the most vulnerable, and positive reveal more pronounced disparities among the least vulnerable [28,29]. We also run some sensitivity analyses to explore potential stratification and subgroup analyses, aiming to evaluate the consistency of the overall results. All analyses were performed in Stata using the *svy* command to account for the primary sampling unit, version 17.1.

## Results

The sociodemographic characteristics of the analytical sample (N = 74,077) of eligible participants are detailed in Table 1. Most participants were women (59.5%), self-reported their race and ethnicity as Brown (44.8%), and completed high school (37.5%). Approximately half of the sample either lacked knowledge or believe that chloroquine has a protective effect against COVID-19. Overall, 20.4% (95% CI: 20.0; 20.7) of participants reported believing that chloroquine was effective, while 27.2% (95% CI: 26.7; 27.7) indicated that they did not know.

The highest prevalence of a lack of knowledge and belief in the protective effect of chloroquine was observed in participants who identified as Indigenous. Participants with university degrees had a lower percentage of belief in the effect of chloroquine (16.7%) compared to those with lower levels of education. A gradient in education levels and wealth quartiles was noted among those unaware of the actual effect of chloroquine. The prevalence of a lack of knowledge regarding chloroquine's protective effect was higher among individuals with lower education levels and those in the lowest wealth quartile. Conversely, among individuals who believe in the protective effect of chloroquine, the highest prevalence was observed in the lowest socioeconomic position (Table 1).

The associations between sociodemographic variables and a lack of knowledge, as well as the belief in the protective effect of chloroquine against COVID-19, are presented in Table 2. Women had lower odds of believing in chloroquine's protective effects than men. Marginalized racial and ethnicity groups, along with individuals with lower education levels and socioeconomic status, had higher odds of mistakenly believing that chloroquine prevented COVID-19. Overall, those with lower education and socioeconomic status were more likely than their counterparts to report a lack of knowledge about the protective effects of chloroquine.

The distribution of the jeopardy index showed that 2.9% of the population comprised male, white, highest education, and highest wealth quartile, and 4.1% consisted of women, Black-Brown-East Asian-Indigenous, lowest education, and lowest wealth quartile (S1 Table). In the middle category of the index (4), the proportion of sex and ethnicity was too similar to the overall population (S2 Table).



**Table 1.** Sociodemographic characteristics and beliefs in the protective effects of chloroquine against COVID-19, categorized by sociodemographic variables. Brazil, EPICOVID (rounds 1 to 3), 2020. N=74,077[a].

| Variables | n (%) | Belief in the chloroquine's protective effect | | |
|---|---|---|---|---|
| | | No % (95%CI) | Yes % (95%CI) | Don't know % (95%CI) |
| Sex | | | | |
| Male | 29,986 (40.5) | 51.6 (50.9; 52.3) | 22.7 (22.1; 23.2) | 25.7 (25.1; 26.3) |
| Female | 44,091 (59.5) | 53.0 (52.1; 53.2) | 18.8 (18.3; 19.2) | 28.2 (27.6; 28.8) |
| Race and ethnicity | | | | |
| White | 27,941 (37.7) | 53.6 (52.9; 54.3) | 18.8 (18.3; 19.3) | 27.6 (26.9; 28.3) |
| Brown | 33,198 (44.8) | 51.4 (50.8; 52.0) | 21.8 (21.3; 22.3) | 26.8 (26.2; 27.4) |
| Black | 9,828 (13.3) | 53.8 (52.6; 55.0) | 19.1 (18.3; 20.0) | 27.1 (26.2; 28.0) |
| East Asian | 2,086 (2.8) | 50.1 (48.1; 52.1) | 21.9 (20.0; 23.9) | 28.0 (26.1; 30.0) |
| Indigenous | 1,024 (1.4) | 46.6 (43.4; 49.8) | 25.0 (22.3; 27.9) | 28.4 (25.6; 31.4) |
| Education level | | | | |
| University degree | 16,816 (22.7) | 63.9 (63.1; 64.7) | 16.7 (16.0; 17.3) | 19.4 (18.8; 20.0) |
| High school | 27,812 (37.5) | 53.0 (52.3; 53.7) | 21.1 (20.5; 21.6) | 25.9 (25.3; 26.6) |
| Elementary school | 12,795 (17.3) | 45.4 (44.5; 46.4) | 21.9 (21.2; 22.7) | 32.6 (31.7; 33.6) |
| Incomplete elementary school | 16,654 (22.5) | 45.3 (44.4; 46.3) | 21.7 (21.0; 22.4) | 33.0 (32.0; 34.0) |
| Wealth quartiles | | | | |
| 1 (Richest) | 18,508 (25.0) | 57.9 (57.1; 58.6) | 19.5 (18.9; 20.1) | 22.6 (22.0; 23.3) |
| 2nd | 18,504 (25.0) | 53.9 (53.1; 54.7) | 19.5 (18.9; 20.1) | 26.6 (25.9; 27.4) |
| 3rd | 18,532 (25.0) | 51.3 (50.5; 52.2) | 19.8 (19.3; 20.4) | 28.8 (28.0; 29.7) |
| 4 (Poorest) | 18,533 (25.0) | 46.7 (45.8; 47.6) | 22.6 (21.9; 23.3) | 30.7 (29.9; 31.5) |
| **Total** | **100.0** | **52.4 (52.0; 52.9)** | **20.4 (20.0; 20.7)** | **27.2 (26.7; 27.7)** |

[a]Analytical sample: valid information for the main outcome and main exposure (jeopardy index).

The prevalence of belief and lack of knowledge regarding the protective effect of chloroquine against COVID-19 by the Jeopardy index is presented in the Fig 1. In all groups, most participants did not believe in the protective effect of chloroquine. However, as Jeopardy index scores increased, the proportion of participants who did not believe in the protective effect of chloroquine decreased. Conversely, the percentage of individuals who were unaware of the chloroquine's effect increased as Jeopardy index scores rose, ranging from 18.7% (CI95%: 17.0–20.4) among people in the lowest jeopardy index score to 33.6% (CI95%: 31.7–35.4) among the more vulnerable population. Compared to those in the lowest Jeopardy index score [male, white, highest education level, and highest wealth quartile], individuals with a score of three or higher had greater odds of reporting belief in the chloroquine's protective effects (Table 3). In comparison, participants with a score of two or above had higher odds of being unaware of the effect of chloroquine. The differences between scores were more pronounced among those who were unaware of the chloroquine's effect than among believers, as summarized by the indicators of absolute and relative inequalities (SII and CIX, respectively). By the study round, we observed an increase in the strength of the associations over time (S3 Table). The results were consistent according to alternative jeopardy indexes using different categories of race and ethnicity (S4 Table).

The effective non-pharmacological measures to prevent SARS-CoV-2 infection showed almost universal "yes" responses. No disparities in mask use and stay-at-home practices were observed based on the jeopardy index (S5 Table). Using a binary outcome (reference: no/yes) to measure uncertainty (don't know) showed a similar association slightly weaker than comparing "don't know" with "no" as a reference (S6 Table and Table 3).



**Table 2. Association between sociodemographic characteristics and misbelief in the protective effect of chloroquine against COVID-19. Brazil, EPICOVID (rounds 1 to 3), 2020.**

| Variables | Chloroquine's protective effect (Reference: No) [a] | | | |
| --- | --- | --- | --- | --- |
| | Crude | | Adjusted [b] | |
| | Yes OR (95%CI) | Don't know OR (95%CI) | Yes OR (95%CI) | Don't know OR (95%CI) |
| **Sex** | | | | |
| Male | 1.00 | 1.00 | 1.00 | 1.00 |
| Female | **0.81 (0.78; 0.84)** | **1.07 (1.03; 1.10)** | **0.81 (0.78; 0.85)** | **1.07 (1.04; 1.11)** |
| **Race and ethnicity** | | | | |
| White | 1.00 | 1.00 | 1.00 | 1.00 |
| Black-Brown-East Asian-Indigenous [c] | **1.17 (1.12; 1.23)** | 1.01 (0.97; 1.05) | **1.09 (1.05; 1.14)** | **0.91 (0.88; 0.95)** |
| **Education level** | | | | |
| University degree | 1.00 | 1.00 | 1.00 | 1.00 |
| High school | **1.53 (1.45; 1.62)** | **1.61 (1.53; 1.69)** | **1.48 (1.40; 1.57)** | **1.56 (1.48; 1.65)** |
| Elementary school | **1.86 (1.74; 1.98)** | **2.37 (2.23; 2.51)** | **1.77 (1.65; 1.89)** | **2.25 (2.11; 2.40)** |
| Incomplete elementary school | **1.84 (1.72; 1.96)** | **2.40 (2.26; 2.55)** | **1.73 (1.62; 1.86)** | **2.25 (2.11; 2.40)** |
| **Wealth quartiles** | | | | |
| 1st (wealthiest) | 1.00 | 1.00 | 1.00 | 1.00 |
| 2nd | **1.07 (1.02; 1.13)** | **1.26 (1.20; 1.33)** | 0.95 (0.90; 1.01) | **1.10 (1.04; 1.15)** |
| 3rd | **1.15 (1.09; 1.21)** | **1.44 (1.36; 1.52)** | 0.95 (0.90; 1.00) | **1.13 (1.07; 1.20)** |
| 4th (Poorest) | **1.44 (1.36; 1.52)** | **1.68 (1.60; 1.77)** | **1.11 (1.04; 1.18)** | **1.20 (1.14; 1.27)** |

Statistically significant associations were highlighted in bold.

[a]Multinomial logistic regression model. Odds ratios indicate the odds of reporting "yes" or "I don't know" instead of "No". OR <1 indicates that the participants were less likely, and OR >1 suggests that they were more likely to respond "Yes" or "I don't know", instead of "No", to the question "Do you believe chloroquine offers protection against the coronavirus?"

[b]Mutually adjusted for other variables presented in the table.

## Discussion

The findings indicate that during the first months of the pandemic, a widespread misbelief existed regarding the chloroquine's protective effect against the SARS-CoV-2 virus. One in five Brazilians believed in the use of chloroquine as a protection against the coronavirus, while nearly a quarter were uncertain about its effects. Furthermore, our study showed notable disparities, with a higher prevalence of misbelief, particularly among the most vulnerable population groups. On the other hand, the belief in the protective effect of mask usage and adherence to stay-at-home policies was nearly universal and did not exhibit inequalities based on the jeopardy index. These findings demonstrate the specific sources of denialism in Brazil and their inequitable impact on public perceptions.

The high misbelief in the protective effect of chloroquine against COVID-19 could be attributed to the data collection period, which was carried out between May and June 2020. However, it is important to note that there was a high assertive belief in the protective effect of chloroquine (20% answered "yes") even with results and scientific consensus

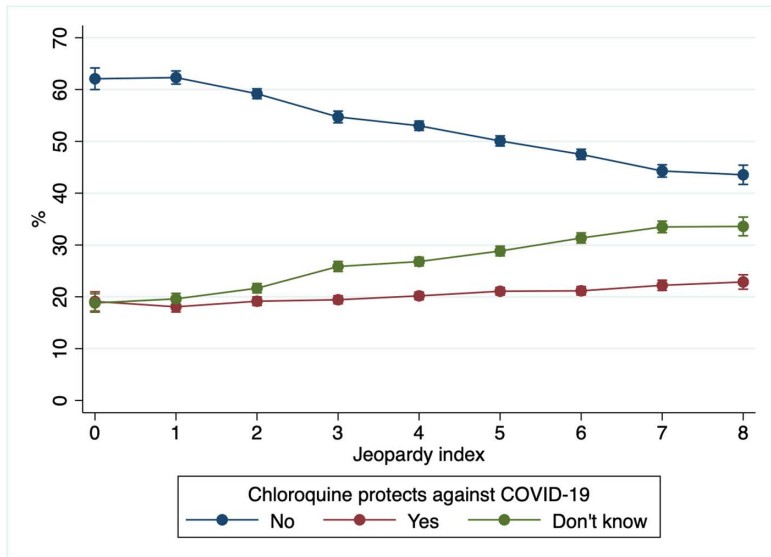

**Fig 1. Predicted prevalence of the belief in the chloroquine's protective effect against COVID-19 according to the jeopardy index. Brazil, EPICOVID (rounds 1 to 3), 2020.** Jeopardy index: Zero = male, White, highest education level, and highest wealth quartile; Eight = woman, Black-Brown-East Asian-Indigenous, lowest education level, and lowest wealth quartile.

**Table 3. Belief in the chloroquine's protective effect against COVID-19 according to the jeopardy index. Brazil, EPICOVID (rounds 1 to 3), 2020.**

| Variables | Chloroquine's protective effect(Reference: No) [a] | |
|---|---|---|
| | **Yes**<br>**OR (CI95%)** | **Don't know**<br>**OR (CI95%)** |
| Jeopardy index [b] | | |
| 0 | Ref | Ref |
| 1 | 0.95 (0.83; 1.09) | 1.05 (0.91; 1.21) |
| 2 | 1.07 (0.94; 1.21) | **1.24 (1.09; 1.42)** |
| 3 | **1.20 (1.06; 1.36)** | **1.59 (1.40; 1.80)** |
| 4 | **1.26 (1.12; 1.43)** | **1.71 (1.52; 1.94)** |
| 5 | **1.36 (1.20; 1.54)** | **1.93 (1.72; 2.17)** |
| 6 | **1.49 (1.32; 1.69)** | **2.26 (1.99; 2.56)** |
| 7 | **1.67 (1.46; 1.91)** | **2.51 (2.21; 2.85)** |
| 8 | **1.72 (1.50; 1.97)** | **2.57 (2.21; 2.99)** |
| SII (95%CI) | **−4.3 (−5.3; −3.2)** | **−15.1 (−16.3; −13.9)** |
| CIX (95%CI) | **−3.3 (−4.1; −2.4)** | **−7.1 (−7.9; −6.4)** |

Statistically significant associations were highlighted in bold.

[a]Multinomial logistic regression model. Odds ratios indicate the odds of reporting "yes" or "I don't know" instead of "No". OR <1 indicates that the participants were less likely, and OR >1 suggests that they were more likely to respond "Yes" or "I don't know", instead of "No", to the question "Do you believe chloroquine offers protection against the coronavirus?"

[b]Jeopardy index: Zero = male, White, highest education level, and highest wealth quartile; Eight = woman, Black-Brown-East Asian-Indigenous, lowest education level, and lowest wealth quartile.

[c]Slope index of inequality (SII) and Concentration Index (CIX) (values between −100 and 100) according to the jeopardy index.

described in the first half of 2020 already pointing out the lack of robust evidence on the benefit of chloroquine and warned of its side effects. Since April 2020, the preliminary results of the CloroCovid-19 study have highlighted the inefficacy and risks associated with chloroquine in treating the disease [30]. And our results showed an increase in the prevalence of misbelief over time (from May to the end of June). If the beginning of the pandemic were the cause of the findings, a decrease over time would be expected.

In addition, the most current evidence indicates a similar percentage of the Brazilian population that continues to believe in the protective effect of chloroquine. The survey "The Face of Democracy", for example, identified, in April 2021, that one in four Brazilians claims to have used drugs of the so-called "early treatment" against COVID-19 or to prevent infection by the SARS-CoV-2 virus.

The executive power's milestone in the public defense of chloroquine/hydroxychloroquine occurred on March 27, 2020, with a post by the President at that time on a social media platform. The post cited "accurate information" about chloroquine having a "high success rate" referencing a pre-print paper (published only in July 2020) that evaluated the effect of hydroxychloroquine in a non-randomized study with 36 French patients, following another post published at the time [31]. In February 2020, a parliamentary commission was established to address the COVID-19 crisis. Since then, technical documents and positions of the Federal Council of Medicine have been published supporting chloroquine mainly in the so-called "early treatment", even for cases with mild symptoms, as observed in an Informative Note from May 2020. Since then, strategies such as the COVID Kit have been popularized, including the use of federal government digital platforms (*TrateCOV*) [31]. In this context, more robust scientific evidence has been signaling the null effect of chloroquine in the treatment and prevention of COVID-19, in addition to reports of side effects that were widely recorded throughout the pandemic [32–34].

The US Food and Drug Administration (FDA) stated in June 2020 that chloroquine and hydroxychloroquine are ineffective in treating SARS-CoV-2 infections. During the same period, the World Health Organization announced the interruption of clinical trials using chloroquine. In Brazil, except for the Federal Council of Medicine, scientific societies, scientists, and health councils warned about the risks of using chloroquine to face the pandemic. In May 2020, the National Health Council published a note indicating that scientific evidence suggested side effects without any benefit for outcomes related to COVID-19 [35]. Nevertheless, the defense of the drug by the president of the republic and his supporters continued throughout 2020 and 2021 [13,36]. Still, in 2022, statements about the benefits of chloroquine were made in several public interviews conducted by the head of the executive power [36].

The ineffectiveness of chloroquine (or hydroxychloroquine) in combating the SARS-CoV-2 virus is more than consolidated [37–39]. Findings from the DETECTCOV-19 cohort in Manaus indicate a higher seroprevalence (immunoglobulin G positivity) among individuals who self-medicated as a prophylactic strategy [40]. Similar results were observed in studies on the effect of hydroxychloroquine as a preventive or post-exposure therapy [41]. However, the population's perception of chloroquine's benefits (and others, such as ivermectin) persists, according to results from 2021 and 2022 [23,42,43].

Scientific denialism as a government policy may have caused a greater concentration of misperception about the role of chloroquine in facing the pandemic. Our findings identify inequalities, especially for individuals who were uncertain whether the drug protected them against the SARS-CoV-2 virus. Although the answer "I do not know" may seem to be a lack of knowledge, it represents the population's doubt influenced by the president's speeches. It is noteworthy that for effective non-pharmacological measures to prevent infection (staying at home and wearing a mask), the percentage of respondents who did not know was < 1%, respectively. Thus, lack of knowledge seems to be more influenced by the government's denialist strategy than by a scientific question about the drug's effect. Therefore, it is plausible to understand that the Brazilian federal government's strategy was an initiative to discredit the scientific consensus and create distrust among the population, making it challenging to adopt effective strategies to face the pandemic [12,44–47]. The perception of the available treatments tends to reduce the adoption of preventive measures. The governmental strategy aimed to

achieve herd immunity (i.e., "greater contamination possible") [13,48] to achieve economic goals unrelated to fundamental human rights, prioritizing the protection of life. The herd immunity proved impractical as was speculated by science since 2020, the chloroquine treatment was ineffective, and the result was high mortality attributable to the federal government's denialist policy of the federal government [4,49].

The risk of political denialism interfering with public health is a growing concern. It is not new, but it is affecting more countries across different areas, even before the pandemic [50]. Considering our historical inequalities in health and the previous evidence aligned with the present findings, political denialism has the potential to eliminate societal achievements in population health in Brazil and exacerbate the marked and unfair inequities. Effective public health approaches can reduce the burden of disease by addressing macro-level injustices and improving population health, benefiting all human beings. As stated by Victora et al. [8], "the challenge is ultimately political", even if it will be for facing the political denialism in public health. Several characteristics can play an important role in the effects of denialism on health, such as media exposure and regional context [50], which could be evaluated in future studies. In addition to the study limitations related to sampling and sample characteristics [20,21], a possible limitation of this analysis is the high percentage of losses and refusals (46−47%), which can lead to a differential bias. The refusal might be higher among wealthier people who believe in the chloroquine effect [51,52]. However, this possible bias may produce an underestimation of the misbelief in chloroquine's protective effect against the SARS-CoV-2 virus. Thus, the potential selection bias may overestimate the association but simultaneously decrease the magnitude of the misbelief in the Brazilian population's perceived effect of chloroquine.

## Conclusions

The misbelief in chloroquine's protective effect was widespread and affected more marginalized social groups among the Brazilian population. People with greater social vulnerability were those most affected by a lack of knowledge about the effectiveness of chloroquine. The denialist federal management of the pandemic in Brazil seems to produce inequitable misbelief in chloroquine's protective effect, which may have influenced the tragic mortality observed in the country.

## Supporting information

**S1 Table. Jeopardy index distribution. Brazil, EPICOVID (rounds 1–3), 2020.** [a]Jeopardy index: Zero = male, White, highest education level, and highest wealth quartile; Eight = woman, Black-Brown-East Asian-Indigenous, lowest education level, and lowest wealth quartile.
(DOCX)

**S2 Table. Study's sample description according to the jeopardy index. Brazil, EPICOVID (rounds 1–3), 2020.**
[a]Jeopardy index: Zero = male, White, highest education level, and highest wealth quartile; Eight = woman, Black-Brown-East Asian-Indigenous, lowest education level, and lowest wealth quartile.
(DOCX)

**S3 Table. Belief in the chloroquine's protective effect against COVID-19 according to the jeopardy index by the study's rounds. Brazil, EPICOVID, 2020.** Statistically significant associations were highlighted in bold. [a] Multinomial logistic regression model. Odds ratios indicate the odds of reporting "yes" or "I don't know" instead of "No". OR <1 indicates that the participants were less likely, and OR >1 suggests that they were more likely to respond "Yes" or "I don't know", instead of "No", to the question "Do you believe chloroquine offers protection against the coronavirus?" [b] Jeopardy index: Zero = male, White, highest education level, and highest wealth quartile; Eight = woman, Black-Brown-East Asian-Asian-Indigenous, lowest education level, and lowest wealth quartile. [c] Slope index of inequality (SII) and Concentration Index (CIX) (values between −100 and 100) according to the jeopardy index.
(DOCX)



**S4 Table. Belief in the chloroquine's protective effect against COVID-19 according to alternative jeopardy indexes by removing specific race and ethnicity categories. Brazil, EPICOVID (rounds 1–3), 2020.** Statistically significant associations were highlighted in bold. [a] Jeopardy index: Zero = male, White, highest education level, and highest wealth quartile; Eight = woman, Black-Brown-Indigenous, lowest education level, and lowest wealth quartile. [b] Jeopardy index: Zero = male, White, highest education level, and highest wealth quartile; Eight = woman, Black-Brown, lowest education level, and lowest wealth quartile. [c] Multinomial logistic regression model. Odds ratios indicate the odds of reporting "yes" or "I don't know" instead of "No". OR <1 indicates that the participants were less likely, and OR >1 suggests that they were more likely to respond "Yes" or "I don't know", instead of "No", to the question "Do you believe chloroquine offers protection against the coronavirus?".
(DOCX)

**S5 Table. Prevalence of effective non-pharmacological measures to prevent SARS-CoV-2 infection according to the jeopardy index. Brazil, EPICOVID (rounds 1–3), 2020.** Jeopardy index: Zero = male, White, highest education level, and lowest wealth quartile; Eight = woman, Black-Brown-East Asian-Indigenous, lowest education level, and lowest wealth quartile.
(DOCX)

**S6 Table. Uncertainty about the belief in the chloroquine's protective effect against COVID-19 according to the jeopardy index. Brazil, EPICOVID (rounds 1–3), 2020.** Statistically significant associations were highlighted in bold. [a] Jeopardy index: Zero = male, White, highest education level, and highest wealth quartile; Eight = woman, Black-Brown-East Asian-Indigenous, lowest education level, and lowest wealth quartile. [b] Logistic regression model. Odds ratios indicate the odds of reporting "I don't know" compared to "No/Yes". OR <1 indicates that the participants were less likely, and OR >1 suggests that they were more likely to respond "I don't know" instead of "Yes or No", to the question "Do you believe chloroquine offers protection against the coronavirus?".
(DOCX)

## Author contributions

**Conceptualization:** Bruno Pereira Nunes, Inácio Crochemore-Silva, Grégore I. Mielke, Luis Paulo Vidaletti, Mariangela Freitas da Silveira, Pedro C. Hallal.

**Data curation:** Luis Paulo Vidaletti, Marilia Arndt Mesenburg, Mariangela Freitas da Silveira.

**Formal analysis:** Bruno Pereira Nunes, Grégore I. Mielke, Luis Paulo Vidaletti.

**Funding acquisition:** Luis Paulo Vidaletti, Mariangela Freitas da Silveira, Pedro C. Hallal.

**Investigation:** Inácio Crochemore-Silva, Marilia Arndt Mesenburg, Mariangela Freitas da Silveira, Pedro C. Hallal.

**Methodology:** Bruno Pereira Nunes, Inácio Crochemore-Silva, Grégore I. Mielke, Luis Paulo Vidaletti, Marilia Arndt Mesenburg, Mariangela Freitas da Silveira, Pedro C. Hallal.

**Project administration:** Marilia Arndt Mesenburg, Mariangela Freitas da Silveira, Pedro C. Hallal.

**Resources:** Mariangela Freitas da Silveira, Pedro C. Hallal.

**Supervision:** Pedro C. Hallal.

**Validation:** Grégore I. Mielke, Pedro C. Hallal.

**Visualization:** Bruno Pereira Nunes, Grégore I Mielke, Luis Paulo Vidaletti, Pedro C. Hallal.

**Writing – original draft:** Bruno Pereira Nunes, Inácio Crochemore-Silva, Grégore I. Mielke, Pedro C. Hallal.

**Writing – review & editing:** Bruno Pereira Nunes, Inácio Crochemore-Silva, Grégore I. Mielke, Luis Paulo Vidaletti, Marilia Arndt Mesenburg, Mariangela Freitas da Silveira, Pedro C. Hallal.



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
