## [Decision Letter · Decision Letter 0]

14 Sep 2025

Dear Dr. Nunes,

Thank you for submitting your manuscript to PLOS ONE. After careful consideration, we feel that it has merit but does not fully meet PLOS ONE’s publication criteria as it currently stands. Therefore, we invite you to submit a revised version of the manuscript that addresses the points raised during the review process.

We look forward to receiving your revised manuscript.

Kind regards,

Ivan Filipe de Almeida Lopes Fernandes, Ph.D.

Academic Editor

PLOS ONE

Journal Requirements:

For additional information about PLOS ONE ethical requirements for human subjects research, please refer to http://journals.plos.org/plosone/s/submission-guidelines#loc-human-subjects-research .

4. We note that your Data Availability Statement is currently as follows: “The EPICOVID19 datasets are freely available online”

Reviewers' comments:

Reviewer's Responses to Questions

**Comments to the Author**

1. Is the manuscript technically sound, and do the data support the conclusions?

Reviewer #1: Yes

Reviewer #2: Yes

Reviewer #3: Yes

2. Has the statistical analysis been performed appropriately and rigorously?

Reviewer #1: Yes

Reviewer #2: Yes

Reviewer #3: Yes

3. Have the authors made all data underlying the findings in their manuscript fully available?

Reviewer #1: Yes

Reviewer #2: Yes

Reviewer #3: Yes

4. Is the manuscript presented in an intelligible fashion and written in standard English?

Reviewer #1: Yes

Reviewer #2: Yes

Reviewer #3: Yes

Reviewer #1: Reviewer's report

Thank you for the opportunity to review the manuscript titled “Social inequalities in the misbelief of chloroquine’s protective effect against COVID-19: results from the EPICOVID-19 study in Brazil”. This manuscript presents original research examining the socioeconomic inequalities in beliefs regarding chloroquine’s protective effect against COVID-19 in Brazil, using nationally representative data. The topic is timely and relevant, especially in the context of health misinformation, political influence, and public trust during health crises. The study addresses important public health and equity concerns and employs appropriate methodological approaches, including the use of the Slope Index of Inequality (SII) and Concentration Index (CIX).

However, there are several areas that could benefit from further clarification and refinement:

The abstract and introduction are clearly written and provide a coherent overview of the research problem, objectives, and context. The background is well-situated in relevant literature, and the rationale for the study is well-articulated. The authors successfully convey the public health relevance of the research question, and the writing is concise and accessible.

Methodology

1. The manuscript introduces a Jeopardy Index based on the aggregation of four sociodemographic variables reflecting social privilege. While the operationalization appears reasonable, the authors should briefly explain why this specific index was used along with relevant references supporting its use.

2. This section should clearly state which statistical software was used for data analysis (e.g., Stata, R, SPSS), including the version.

Results

1. In the sentence summarizing participant’s demographic characteristics, report the exact percentages for race and education as well.

2. Line 156 can be rephrased like this for clarity: “Approximately 47.9% either lacked knowledge about or did not believe that chloroquine has a protective effect against COVID-19.”

3. In Table 1, I recommend including the counts (n) alongside the percentages (%) for each category of the variables.

4. In Table 2, the sample size is reported as (n = 88,772) in the heading, but this should be written as (N = 88,772) to reflect that it refers to the total sample. Please ensure that the notation for sample size (i.e., N for total sample, n for subsamples) is used consistently throughout the manuscript, including tables, figures, and text.

5. In the tables even though confidence intervals for inequality measures and other estimates are provided, it does not clearly indicate which variables or associations are statistically significant. Although readers can infer significance from whether the CIs include zero, it would improve clarity if statistically significant results were explicitly indicated, either in the text or in table footnotes.

6. The interpretation of SII and CIX results in the text could be expanded. Also, consider including key effect estimates (e.g., percentages, odds ratios, and confidence intervals) in the main text where relevant, so readers don’t have to rely only on the tables.

Discussion

1. The discussion is bold and well-supported with relevant references. The integration of political context and its influence on public beliefs is especially commendable. However, this section could be strengthened further by expanding the discussion on how socially disadvantaged groups are disproportionately affected in such scenarios. While this is touched upon, giving it more emphasis would enhance the depth and equity focus of the discussion.

Reviewer #2: Dear authors,

Congratulations on this manuscript. I believe it has merit for publication after revisions.

Title

• Short title appears truncated (“protective in Brazil”); suggest “protective effect in Brazil.”

Abstract

• Major: Abstract states “Lack of knowledge was 2.49 times greater among women than among men.” That appears inconsistent with the adjusted models (female vs male OR≈1.06 for “don’t know”). The 2.49 figure corresponds to Jeopardy Index 8 vs 0, not to sex alone; please correct.

• Clarify whether “yes” and “don’t know” are both treated as “denialism” in the abstract; the main text provides the rationale, but readers won’t see it here (See Discussion for justification).

Introduction

• Although I find it important in a personal way for the Brazilian context, several sentences speculate on political alignment and behavior. PLOS ONE prioritizes testable, neutral claims; please tone down speculative language or back it with citations framed cautiously.

• Where you assert widespread emphasis on chloroquine/policy decisions, ensure each claim has a source in text (some are cited later in Discussion). Consider moving part of the policy chronology to the Discussion or Supplement.

Methods

• The sampling is three stage (cities, census tracts, households), but it’s unclear whether analyses accounted for weights, clustering, and stratification (e.g., survey adjusted multinomial models; robust SEs). Please specify and, if not used, justify and consider re estimating with survey design corrections.

• You pool three rounds (May–June 2020). Indicate whether survey round was included (e.g., fixed effects) to address temporal shifts in beliefs, and report any round specific sensitivity analyses.

• Treating “don’t know” as “denialism” is conceptually debatable. You provide a rationale later; please pre specify this classification here and add a sensitivity analysis modeling “don’t know” separately (or as uncertainty), and another collapsing “yes vs not yes.”

About Joepardy index:

• • Clarify the distribution (mean, SD, histogram) and percent in each score (0–8).

• • Justify the coding choices (e.g., women=1 vs men=0; non white=1 vs white=0; ordinal steps for education/wealth). Consider a robustness check using an alternative weighting or PCA based index.

• Explain how you handled race/skin color (grouping “non white” merges Black, mixed, Asian, Indigenous). Consider disaggregated models by group to check for heterogeneity (if possible).

• Report full model terms for the multinomial regressions; confirm absence/presence of multicollinearity among index components when they are also modeled individually.

• Clarify whether city (or region) was included to capture contextual differences.

Results

• Reconcile the sex effect language with the models. The tables show only modest female vs male differences for “don’t know” (OR≈1.06).

• When stating that belief is “~20% across the index,” consider presenting 95% CIs for Figure 1 predictions and/or a test for trend/flatness.

Discussion

• Date inconsistency: You attribute high misbelief partly to data collected “between May and August 2020,” but Methods indicate May 14–21, June 4–7, and June 21–24 (no August). Please correct.

• Normative/political passages read strongly opinionated. Reframe in neutral, evidence based terms and minimize conjecture; ensure all assertions are cited and kept proportional to your data (e.g., claims about governmental strategy and herd immunity).

• Since “don’t know” drives the strongest inequalities, emphasize implications for risk communication and health literacy (who, where, how to target), and consider whether media exposure or regional context could mediate associations.

• Where you mention mask/stay at home results, add a brief Supplementary Table rather than “data not shown.”

Limitations

• Explicitly discuss design effects and any post stratification weighting (if used).

• Clarify misclassification risks in self reported beliefs and in grouping heterogeneous racial categories as “non white.”

Conclusions

Avoid causal language; emphasize that findings are associational given cross sectional design.

Reviewer #3: The manuscript addresses a topic of clear relevance to public health, as it discusses a form of social vulnerability across multiple contexts. The study demonstrates methodological rigor and adheres to established ethical standards. I consider it suitable for publication.

**Do you want your identity to be public for this peer review?** For information about this choice, including consent withdrawal, please see our Privacy Policy

Reviewer #1: No

Reviewer #2: **Yes:** JOSE FIRMINO DE SOUSA FILHO

Reviewer #3: No

---

## [Author Response · Author response to Decision Letter 1]

10 Dec 2025

Dear Ivan Filipe de Almeida Lopes Fernandes, Ph.D., Academic Editor, PLOS ONE

We appreciate the opportunity to submit a revised version of our paper. The comments from the reviewers and the editorial team were important in giving us the chance to provide a more interesting and improved version.

We reviewed the points highlighted in the journal requirements section and have made adjustments to all of them.

While addressing the comments, we revised our analytical sample and excluded children and adolescents, who were initially included. Given that the outcome was attributed to the responsibility of children and adolescents, and that the main exposure (jeopardy) includes information from children and adolescents (sex and race), as well as from adults and households (education level and socioeconomic level), we considered that removing children and adolescents from the analytical sample would improve exposure classification and, consequently, strengthen the methodological rigor. Importantly, the updated results are similar, and the interpretation of the findings did not change.

Thank you.

Best wishes,

Bruno P Nunes on behalf of the co-authors

Point-by-point authors’ responses

Journal Requirements:

https://journals.plos.org/plosone/s/file?id=wjVg/PLOSOne_formatting_sample_main_body.pdfand
https://journals.plos.org/plosone/s/file?id=ba62/PLOSOne_formatting_sample_title_authors_affiliations.pdf

Author’s response: We adjusted it. Thank you.

Author’s response: We adjusted it. Thank you.

Author’s response: We added the link to the datasets, dictionaries, and report. Thank you.

Author’s response: Thank you. Answered in the last question.

Reviewers comments:

Reviewer #1: Reviewer's report

Thank you for the opportunity to review the manuscript titled “Social inequalities in the misbelief of chloroquine’s protective effect against COVID-19: results from the EPICOVID-19 study in Brazil”. This manuscript presents original research examining the socioeconomic inequalities in beliefs regarding chloroquine’s protective effect against COVID-19 in Brazil, using nationally representative data. The topic is timely and relevant, especially in the context of health misinformation, political influence, and public trust during health crises. The study addresses important public health and equity concerns and employs appropriate methodological approaches, including the use of the Slope Index of Inequality (SII) and Concentration Index (CIX).

However, there are several areas that could benefit from further clarification and refinement:

The abstract and introduction are clearly written and provide a coherent overview of the research problem, objectives, and context. The background is well-situated in relevant literature, and the rationale for the study is well-articulated. The authors successfully convey the public health relevance of the research question, and the writing is concise and accessible.

Author’s response: thank you.

Methodology

1. The manuscript introduces a Jeopardy Index based on the aggregation of four sociodemographic variables reflecting social privilege. While the operationalization appears reasonable, the authors should briefly explain why this specific index was used along with relevant references supporting its use.

Author’s response: Thank you for the comment. This specific index was used based on the assumption of a multiplicative relationship among sociodemographic variables and health outcomes. We developed and added a reference in the introduction (lines 88-90, tracked changes version) to support the use of the index, which is based on intersectionality theory and similar to other papers.

2. This section should clearly state which statistical software was used for data analysis (e.g., Stata, R, SPSS), including the version.

Author’s response: Thank you. I added the information: “All analyses were performed in Stata using the svy command to account for the primary sampling unit, version 17.1” (lines 163-164, tracked changes version)

Results

1. In the sentence summarizing participant’s demographic characteristics, report the exact percentages for race and education as well.

Author’s response: Great. We now report the exact percentages.

2. Line 156 can be rephrased like this for clarity: “Approximately 47.9% either lacked knowledge about or did not believe that chloroquine has a protective effect against COVID-19.”

Author’s response: Excellent. We adjusted as suggested.

3. In Table 1, I recommend including the counts (n) alongside the percentages (%) for each category of the variables.

Author’s response: We added them. Thanks.

4. In Table 2, the sample size is reported as (n = 88,772) in the heading, but this should be written as (N = 88,772) to reflect that it refers to the total sample. Please ensure that the notation for sample size (i.e., N for total sample, n for subsamples) is used consistently throughout the manuscript, including tables, figures, and text.

Author’s response: Thank you. We changed it in Table 2 and reviewed all the notation for the sample size.

5. In the tables even though confidence intervals for inequality measures and other estimates are provided, it does not clearly indicate which variables or associations are statistically significant. Although readers can infer significance from whether the CIs include zero, it would improve clarity if statistically significant results were explicitly indicated, either in the text or in table footnotes.

Author’s response: Great suggestion. The statistically significant associations were highlighted in bold.

6. The interpretation of SII and CIX results in the text could be expanded. Also, consider including key effect estimates (e.g., percentages, odds ratios, and confidence intervals) in the main text where relevant, so readers don’t have to rely only on the tables.

Author’s response: We see your point. We reviewed the text to provide more auto-explanatory content.

Discussion

1. The discussion is bold and well-supported with relevant references. The integration of political context and its influence on public beliefs is especially commendable. However, this section could be strengthened further by expanding the discussion on how socially disadvantaged groups are disproportionately affected in such scenarios. While this is touched upon, giving it more emphasis would enhance the depth and equity focus of the discussion.

Author’s response: Thank you. We appreciated the comment and edited some points in the discussion section to consider it. Also, we added one paragraph (line 335, , tracked changes version) to enhance the discussion of equity related to denialism.

Reviewer #2: Dear authors,

Congratulations on this manuscript. I believe it has merit for publication after revisions.

Author’s response: Thank you.

Title

• Short title appears truncated (“protective in Brazil”); suggest “protective effect in Brazil.”

Author’s response: Thank you. We corrected it.

Abstract

• Major: Abstract states “Lack of knowledge was 2.49 times greater among women than among men.” That appears inconsistent with the adjusted models (female vs male OR≈1.06 for “don’t know”). The 2.49 figure corresponds to Jeopardy Index 8 vs 0, not to sex alone; please correct.

Author’s response: Thank you so much. We corrected it.

• Clarify whether “yes” and “don’t know” are both treated as “denialism” in the abstract; the main text provides the rationale, but readers won’t see it here (See Discussion for justification).

Introduction

Author’s response: We added it in the abstract.

• Although I find it important in a personal way for the Brazilian context, several sentences speculate on political alignment and behavior. PLOS ONE prioritizes testable, neutral claims; please tone down speculative language or back it with citations framed cautiously.

Author’s response: excellent. We removed all speculative sentences from the paper.

• Where you assert widespread emphasis on chloroquine/policy decisions, ensure each claim has a source in text (some are cited later in Discussion). Consider moving part of the policy chronology to the Discussion or Supplement.

Author’s response: excellent. We removed all speculative sentences from the paper and adjusted the text accordingly.

Methods

• The sampling is three stage (cities, census tracts, households), but it’s unclear whether analyses accounted for weights, clustering, and stratification (e.g., survey adjusted multinomial models; robust SEs). Please specify and, if not used, justify and consider re estimating with survey design corrections.

Author’s response: thank you. We added the following sentence: “All analyses were performed in Stata using the svy command to account for the primary sampling unit, version 17.1” (lines 163-164, tracked changes version)

• You pool three rounds (May–June 2020). Indicate whether survey round was included (e.g., fixed effects) to address temporal shifts in beliefs, and report any round specific sensitivity analyses.

Author’s response: good point. We run sensitivity analysis and present it as Supplementary material. The analysis by the study’s round is described in S Table 3. We added information about it in the methods section (lines xx, tracked version), results (lines xx, tracked version), and discussion (lines xx, tracked version). The sensitivity analysis showed a stronger association over time, reinforcing our findings.

• Treating “don’t know” as “denialism” is conceptually debatable. You provide a rationale later; please pre specify this classification here and add a sensitivity analysis modeling “don’t know” separately (or as uncertainty), and another collapsing “yes vs not yes.”

Author’s response: The suggestion is quite interesting. However, in our opinion, we did this analysis using multinomial regression. All analyses showed the outcome in three categories: comparing yes vs no, and don’t know vs no. The reference is always the “no” category. If we misunderstood this point, please let us know about that so we can proceed to consider it. Also, we provide a S Table 6 showing the results of an alternative outcome where 0=”yes/no” and 1=”don’t know”. The findings showed a similar pattern with a slightly weaker strength of association compared to “don’t know” vs “no”.

About Joepardy index:

• Clarify the distribution (mean, SD, histogram) and percent in each score (0–8).

Author’s response: Thank you. We provided this table as supplementary material (S Table 1) and in the results section.

• Justify the coding choices (e.g., women=1 vs men=0; non white=1 vs white=0; ordinal steps for education/wealth). Consider a robustness check using an alternative weighting or PCA based index.

Author’s response: Thank you. We provided a supplementary table (S Table 4) showing results from alternative weightings of the race/ethnicity variable in the jeopardy index.

• Explain how you handled race/skin color (grouping “non white” merges Black, mixed, Asian, Indigenous). Consider disaggregated models by group to check for heterogeneity (if possible).

Author’s response: Thank you. We provided a supplementary table (S Table 4) showing results from alternative weightings of the race/ethnicity variable in the jeopardy index. Also, we provided the distribution of the variables (sex, race/ethnicity, education level, and socioeconomic level) according to the jeopardy index (S Table 2)

• Report full model terms for the multinomial regressions; confirm absence/presence of multicollinearity among index components when they are also modeled individually.

Author’s response: Thank you. There is no evidence of multicollinearity. The highest correlation is between education level and wealth index (0.4173).

• Clarify whether city (or region) was included to capture contextual differences.

Author’s response: Region was not included in the analysis.

Results

• Reconcile the sex effect language with the models. The tables show only modest female vs male differences for “don’t know” (OR≈1.06).

Author’s response: thank you. We agreed and removed this affirmation (line 194, tracked version).

• When stating that belief is “~20% across the index,” consider presenting 95% CIs for Figure 1 predictions and/or a test for trend/flatness.

Author’s response: Thank you. We added the 95% CIs to the text. Figure 1 already presents the 95% CI.

Discussion

• Date inconsistency: You attribute high misbelief partly to data collected “between May and August 2020,” but Methods indicate May 14–21, June 4–7, and June 21–24 (no August). Please correct.

Author’s response: We appreciate your comment. We adjusted.

• Normative/political passages read strongly opinionated. Reframe in neutral, evidence based terms and minimize conjecture; ensure all assertions are cited and kept proportional to your data (e.g., claims about governmental strategy and herd immunity).

Author’s response: excellent. We removed all speculative sentences from the paper.

• Since “don’t know” drives the strongest inequalities, emphasize implications for risk communication and health literacy (who, where, how to target), and consider whether media exposure or regional context could mediate associations.

Author’s response: Thank you. We appreciated the comment and edited some points in the discussion section to consider it. Also, we added one paragraph (line 335) to enhance the discussion of equity related to denialism, media exposure, and regional context.

• Where you mention mask/stay at home results, add a brief Supplementary Table rather than “data not shown.”

Author’s response: Thank you. We added them as supplementary material.

Limitations

• Explicitly discuss design effects and any post stratification weighting (if used).

Author’s response: We added the information in the methods section. We used primary sampling unit to consider the design of the study.

• Clarify misclassification risks in self reported beliefs and in grouping heterogeneous racial categories as “non white.”

Author’s response: Thank you. We did supplementary materials (S Table 4) to show alternative jeopardy indexes considering race/ethnicity. The findings were virtually equal to the primary analysis.

Conclusions

Avoid causal language; emphasize that findings are associational given cross

---

## [Decision Letter · Decision Letter 1]

11 Jan 2026

Social inequalities in the misbelief of chloroquine’s protective effect against COVID-19: results from the EPICOVID-19 study in Brazil

PONE-D-25-30857R1

Dear Dr. Nunes Pereira,

We’re pleased to inform you that your manuscript has been judged scientifically suitable for publication and will be formally accepted for publication once it meets all outstanding technical requirements.

Kind regards,

Ivan Filipe de Almeida Lopes Fernandes, Ph.D.

Academic Editor

PLOS One

Additional Editor Comments (optional):

Reviewers' comments:

Reviewer's Responses to Questions

**Comments to the Author**

Reviewer #1: All comments have been addressed

Reviewer #2: All comments have been addressed

2. Is the manuscript technically sound, and do the data support the conclusions?

Reviewer #1: Yes

Reviewer #2: Yes

3. Has the statistical analysis been performed appropriately and rigorously?

Reviewer #1: Yes

Reviewer #2: Yes

4. Have the authors made all data underlying the findings in their manuscript fully available?

Reviewer #1: Yes

Reviewer #2: No

5. Is the manuscript presented in an intelligible fashion and written in standard English?

Reviewer #1: Yes

Reviewer #2: Yes

Reviewer #1: I have reviewed the revised manuscript and find that the authors have addressed the reviewer comments adequately. The revisions have strengthened the manuscript, particularly in terms of methodological clarity and interpretation of results. I do not have any further concerns and recommend the manuscript for publication.

Reviewer #2: Dear authors, congratulations on your work; I believe it is acceptable for publication in its current format.

**Do you want your identity to be public for this peer review?** For information about this choice, including consent withdrawal, please see our Privacy Policy

Reviewer #1: No

Reviewer #2: **Yes:** José Firmino de Sousa Filho

---

## [Editor Report · Acceptance letter]

PONE-D-25-30857R1

PLOS One

Dear Dr. Nunes,

I'm pleased to inform you that your manuscript has been deemed suitable for publication in PLOS One. Congratulations! Your manuscript is now being handed over to our production team.

Kind regards,

on behalf of

Dr. Ivan Filipe de Almeida Lopes Fernandes

Academic Editor

PLOS One